# How does Architecture Influence the Base Capabilities of Pre-trained Language Models? A Case Study Based on FFN-Wider and MoE Transformers

**Xin Lu**[1], **Yanyan Zhao**[1,*] **Bing Qin**[1] , **Liangyu Huo**[2] , **Qing Yang**[2] , **Dongliang Xu**[2]

[1]Research Center for Social Computing and Information Retrieval, Harbin Institute of Technology
[2]Du Xiaoman (Beijing) Science Technology Co., Ltd.
[1]{xlu, yyzhao, qinb}@ir.hit.edu.cn
[2]{huoliangyu, yangqing, xudongliang}@duxiaoman.com

## Abstract

Pre-trained language models have been proven to possess strong base capabilities, which not only excel in in-distribution language modeling but also show powerful abilities in out-of-distribution language modeling, transfer learning and few-shot learning. Unlike existing work focusing on the influence of scale on base capabilities, our work examines the influence of architecture on those. Specifically, our concern is: How does architecture influence the base capabilities of pre-trained language models? In this work, we attempt to explain and reverse the decline in base capabilities caused by the architecture of FFN-Wider Transformers, seeking to provide some insights. Through analysis, we found the contribution ratio of Multi-Head Attention (a combination function) to pre-trained language modeling is a key factor affecting base capabilities. FFN-Wider Transformers reduce the contribution ratio of this combination function, leading to a decline in base capabilities. We confirmed this by experiments and proposed Combination Enhanced Architecture (CEA) to address the decline in base capabilities of such models. Significantly, we extended our explanation and CEA to Mixture of Experts (MoE) Transformers. We successfully achieved significant improvements in base capabilities on a 14B parameter MoE model, demonstrating the practical application value of our work. This also indicates that our analysis has a certain guiding significance for architecture analysis, architecture improvement and architecture design.

## 1   Introduction

Recent research has discovered that pre-trained language models possess strong base capabilities [24, 7, 3, 22]. They can not only address in-distribution language modeling which is usually their pre-training objective, but also unexpectedly excel in **out-of-distribution language modeling**, **transfer learning**, **few-shot learning**, etc. This has attracted the attention of many researchers.

However, it has also been observed that the cost of pre-training a language model is substantial, with the trial-and-error approach based on empirical improvements proving to be very expensive. Consequently, there is a desire to gain insights into the final performance by analyzing factors like scale and architecture that directly determine the base capabilities of models.

In this process, much attention has been focused on analyzing scale, leading to the formulation of compelling **scaling laws** [14, 11] that drive the trend of enhancing base capability by increasing parameter numbers, data volume and training tokens. In contrast, the impact of architecture has not

---

*Email corresponding.

38th Conference on Neural Information Processing Systems (NeurIPS 2024).

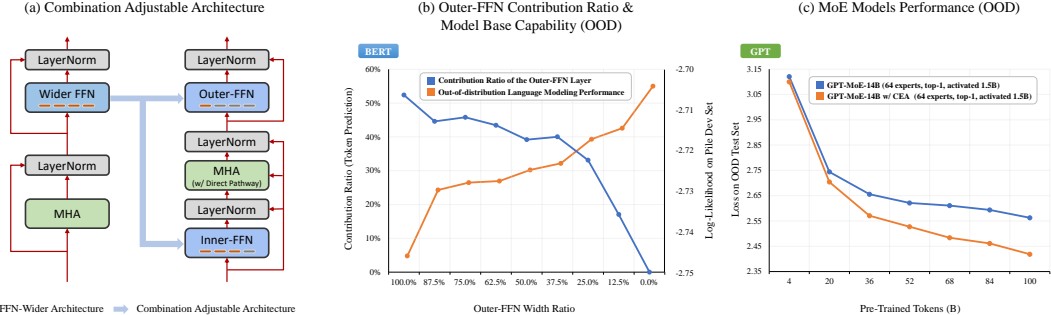

Figure 1: Illustration showing that: 1) the synchronous improvement in model base capability as the contribution ratio of the Outer-FFN layer (a transformation function) decreases, that is, the contribution ratio of the MHA layer (a combination function) increases. This reveals a key factor affecting model's base capabilities. 2) Combination Enhanced Architecture (CEA) was designed based on this factor and applied to MoE models, resulting in an improvement in base capability.

received sufficient attention. According to the basic principle of inductive bias in machine learning, **model architecture** is also a crucial factor affecting base capabilities, and its impact could be equally decisive.

Some studies have already noted the significant influence of architecture on the base capabilities. For instance, [31] have observed considerable differences in base capabilities among various Transformer architecture variants. Some variants, though larger in scale and more powerful in pre-training performance than vanilla Transformers, exhibit significantly reduced performance in downstream tasks. **This suggests different architecture models vary greatly in converting pre-training performance into base capabilities.** Simply increasing scale does not resolve all issues, and exploring the impact of architecture on base capabilities is crucial.

However, despite these studies demonstrating the key influences of model architecture on base capabilities, the understanding of the underlying mechanisms of these influences remains limited. In this work, we attempt to explain the base capability change caused by a specific model architecture change and identify the underlying influencing factor, then design experiments to validate our explanation and influencing factor, and finally propose a generalizable enhancement method.

Specifically, **we first focus on FFN-Wider Transformers**. This architecture has wider FFN layers, but we found that the change leads to a significant decrease in base capabilities compared to vanilla Transformers. As shown in Figure 2, under similar pre-training performance, the FFN-Wider models exhibit a noticeable decline in base capabilities compared to the vanilla models. We believe such a simple change, leading to significant differences in base capabilities, makes for a good object of study in exploring how architecture impacts base capabilities.

Then, we attempted to explain this change in base capabilities. Through analysis, we concluded that the MHA (Multi-Head Attention) layer in the Transformer is a combination function, and the FFN (Feed-Forward Network) layer is a transformation function, with the former being a focused expression of the combinability of language. During their contribution to pre-trained language modeling, **the actual contribution ratio of the MHA layer, as a combination function, is a key factor affecting the model's base capabilities**. The FFN-Wider Transformer models directly enhance the FFN layer, indirectly reducing the combination function's actual contribution ratio to pre-trained language modeling, thereby leading to a significant decline in base capabilities.

To validate our explanation for it, we designed a Combination Adjustable Architecture (CAA), as depicted in Figure 1(a). This architecture bifurcates a wider FFN into two parts with adjustable width ratios: one remains in its original position as a transformation function, known as the Outer-FFN, and the other is relocated within the MHA layer, transformed through a special design into an Inner-FFN that solely enhances the combination function. We controlled the width ratio of the Outer-FFN, reducing it gradually from 100% to 0%. In Figure 1(b), we observed that the actual contribution ratio of the Outer-FFN progressively decreased, indicating a corresponding increase in the actual contribution ratio of the MHA layer. At the same time, we also observed a gradual improvement in base capabilities. These reveal a key phenomenon that confirms our explanation: **as the actual**

**contribution ratio of the MHA layer (a combination function) increases, there is a general synchronous improvement in the model's base capabilities**.

Subsequently, we identified the optimal width ratio of two parts of FFN for the Combination Adjustable Architecture (CAA) and established it as Combination Enhanced Architecture (CEA), and comparing this architecture with the original FFN-Wider Transformer. We conducted experiments on various scales of BERT and GPT models, with all results robustly supporting our explanation.

Importantly, we also noticed that existing work has observed a similar decline in base capabilities in the Mixture of Experts (MoE) Transformers. **We applied our explanation and CEA to MoE Transformers as well**. We pre-trained a 14B parameter GPT architecture MoE model and its improved version with CEA on 100B tokens (using CC and C4 from SlimPajama [29]). With the same number of parameters, computational load, and pre-training steps for both versions, the improved version with CEA showed significant improvements in out-of-distribution language modeling and few-shot learning. The performance of the out-of-distribution language modeling (average performance across all domains in SlimPajama except for CC and C4) is shown in Figure 1(c), which can prove the practical application value of our work.

Overall, the actual contribution ratio of the MHA layer (a combination function) is likely a universal factor affecting model's base capabilities, which can provide valuable insights for architecture analysis, improvement and design.

## 2 Background

Subsequent sections will involve specific analyses and experiments of FFN-Wider and MoE transformers. Therefore, in this section, we introduce the relevant background of base capabilities, as well as the unified evaluation schemes and evaluation tasks.

### 2.1 Base Capabilities

Pre-trained language models not only address in-distribution language modeling but also unexpectedly show strong base capabilities. In this work, we focus on the following base capabilities:

**Out-of-Distribution Language Modeling**    This reflects the out-of-distribution generalization capability of pre-trained language models. Models that learn more essential language features often outperform others, which is a good measure of the base capabilities of pre-trained language models.

**Transfer Learning**    This is a recognized base capability of pre-trained language models. The works proposing GPT [24] and BERT [7] have established the "pre-training and fine-tuning" transfer learning paradigm under the Transformer architecture.

**Few-shot Learning**    This is also a recognized base capability. Some works [25, 3] found large-scale pre-trained language models could complete many NLP tasks with no or only a few demonstrations, revealing their remarkable few-shot learning capabilities.

### 2.2 Evaluation Schemes

In this work, we primarily explore how the architecture of a model influences its base capabilities in pre-trained language models, which requires designing a sound approach for quantitative analysis.

Comparing the base capabilities of one architecture model against another is straightforward: we simply compare their performance on a variety of representative tasks. However, changes in base capabilities cannot be directly attributed to changes in architecture alone, as model capabilities are also influenced by factors like scale.

**This leads to a question**: under what conditions can we compare the base capabilities of two different architecture models and be more confident that the differences are due to architecture variations?

**Our scheme to this question**: we compare the models when they use the same pre-training data and objectives, and have similar levels of pre-training performance (i.e., language modeling performance on an in-distribution development set).

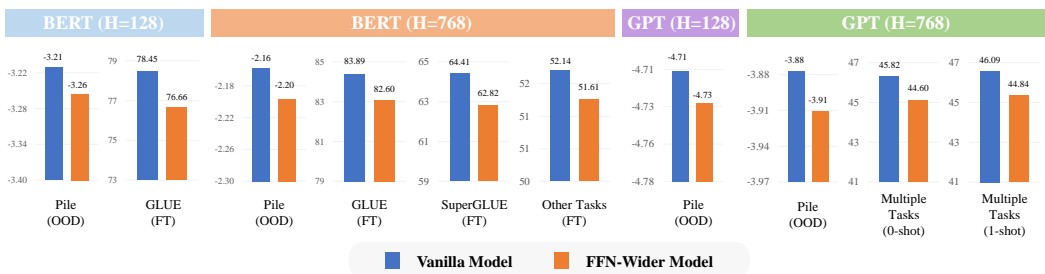

Figure 2: Comparison of the base capabilities between FFN-Wider and Vanilla Transformers.

**The underlying rationale**: two different architecture models, when trained on the same corpus and objectives, both gain base capabilities by reducing the loss in pre-trained language modeling. If one architecture model achieves greater base capabilities with the same reduction in language modeling loss, it is highly likely that this additional benefit stems from the inductive bias of its architecture. In other words, when pre-training performance is aligned, differences in base capabilities of models are likely reflecting the impact of architecture inductive biases.

This scheme is more appropriate for cross-architecture analysis compared to aligning pre-training steps, parameter numbers, or computational load, and we explain it in Appendix A.

Unless specified otherwise, all comparisons of base capabilities across architecture models in this work are conducted under the condition of aligned pre-training performance.

**The experiment with MoE transformers is an exception**, because we need to demonstrate practicality. Therefore, in the comparative experiments between the vanilla MoE model and the improved MoE model, not only is pre-training performance aligned, but pre-training steps, parameter numbers, and computational load are also kept consistent.

## 2.3 Evaluation Tasks

Due to subsequent experiments involving out-of-distribution language modeling evaluation, transfer learning evaluation and few-shot learning evaluation, it is necessary to clarify the selection of pre-training corpus and out-of-distribution test corpus, as well as the selection of downstream tasks. We provide a detailed introduction in Appendix B.

## 3 FFN-Wider Transformers vs. Vanilla Transformers

This work focuses on the base capabilities of FFN-Wider Transformers. A typical Transformer model has Feed-Forward Network (FFN) layers. Assuming the hidden dimension is $d$, the standard intermediate dimension of an FFN layer is $4d$. However, an FFN-Wider Transformer model means the intermediate dimension of its FFN layer exceeds $4d$.

We conducted experiments on various models with the intermediate dimension set to $32d$, aligning pre-training performances with those of vanilla models. The results are shown in Figure 2.

We found that, at the same level of pre-training performance, the Transformer models with wider FFN layers exhibit a noticeable decline in performance on most downstream tasks, indicating a deterioration in their base capabilities. This presents us with a good research object.

## 4 Why FFN-Wider Transformers Have Worse Base Capabilities?

### 4.1 Combination and Transformation

Transformers consist of MHA and FFN layers. Considering a certain position in a sequence, the updated representation obtained after an MHA layer is a **combination** of the entire sequence context; whereas the updated representation obtained after an FFN layer is from a **transformation** of that position representation alone, context-insensitive.

When considering models composed of only one MHA layer or one FFN layer, we find that they can both be used for language modeling. If treated as black boxes focusing only on input and output, there is no difference. However, the way they accomplish language modeling is different, or rather, their inductive biases differ: the FFN layer directly maps the previous token to the target token, representing a one-to-one **transformation function**; while the MHA layer uses the entire sequence to calculate the target token, representing a many-to-one **combination function**. The latter aligns more closely with the combinability of language.

Further considering the multi-layer stacking of MHA and FFN layers in a Transformer model, we easily understand that although all layers contribute to the final language modeling objective, they do so with different inductive biases. Among these, the MHA layer might be the central embodiment of the model architecture's expression of the combinability of language.

With this understanding, when we revisit the scenarios where the FFN layer is widened or its capacity is increased, we have reason to suspect this will lead to a change in the actual contribution ratios of the transformation and combination function, thereby affecting the model's expression of the combinability of language, which is ultimately reflected in changes in basic capabilities.

**Our hypothesis**: the actual contribution ratio of the MHA layer (a combination function) is a key factor affecting the model's base capabilities. The FFN-Wider Transformers directly enhance the FFN layer, indirectly reducing the combination function's actual contribution ratio to pre-trained language modeling, thereby leading to a significant decline in base capabilities.

## 4.2 Contribution Ratio Analysis

To verify our hypothesis, we first need to confirm whether the actual contribution ratio of MHA layers truly decreases, requiring quantitative analysis.

### 4.2.1 Mutual Information

The first method is Mutual Information (MI). For positions in the sequence where a predicted token is to be output, we can focus on the representations of these positions after each layer and calculate the MI between these representations and the target tokens. Then, since we have MI before and after passing through any layer, we can calculate the contribution ratios of the MHA and FFN layers.

We adopted the mutual information estimate method proposed by [32], which mainly involves converting the representations into discrete variables through clustering, and then calculating the mutual information. For the definition of MI and more specific details, please refer to Appendix F.

We analyzed four small-scale models (H=128): a vanilla BERT, an FFN-Wider BERT, a vanilla GPT and an FFN-Wider GPT. The pre-training performances of the vanilla models and the FFN-Wider models are aligned. We plotted the MI results in Figure 3, where the blue, orange and grey lines represent the cumulative MI increment contributions of the MHA layer, FFN layer and Block, respectively. It can be seen that in the FFN-Wider models, the MI contribution of the FFN layer is significantly higher than that in the vanilla models, preliminarily validating our hypothesis.

### 4.2.2 Token Prediction

Since the MI estimate requires clustering a large number of representations and is costly when the hidden dimension is high, we tried another more direct method called Token Prediction (TP). It involves predicting tokens directly from the hidden representations. The approach is as follows:

We still obtained representations from each layer, first dividing them into sets based on the corresponding output token. Then, we normalized all representations within a set and use their mean as the category vector for that token. During this process, we eliminated tokens where the set size was smaller than 50. For the representations of each layer, we then calculated the cosine similarity with the category vectors of tokens in that layer, selecting the token with the highest similarity as the prediction result for that representation. In this way, we could calculate a token prediction accuracy for each layer. The subsequent approach was identical to that of the MI method.

We analyzed four small-scale (H=128) and four large-scale (H=768) models concurrently. We plotted the accuracy increment contribution ratio of the FFN layer for each model, as shown in Figure 4. It

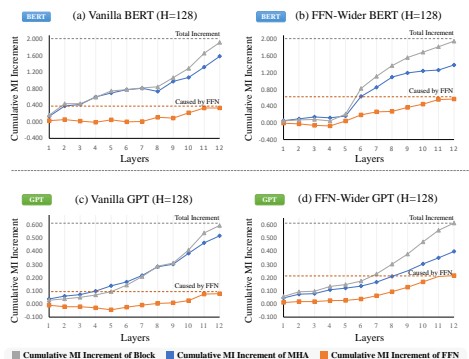

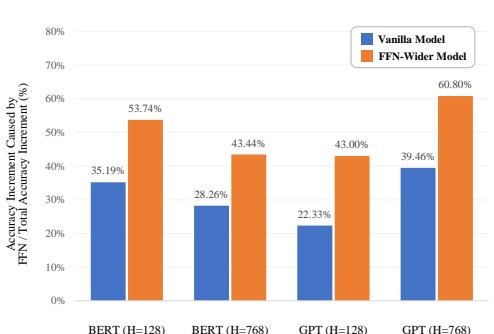

Figure 3: Contribution ratio analysis based on Mutual Information(MI) for various transformers.

Figure 4: Contribution ratio analysis based on Token Prediction (TP) for various transformers.

can be seen that the contribution of FFN layers in FFN-Wider models remains higher than that in vanilla models, reaffirming our hypothesis.

# 5 Combination Adjustable Architecture

Although the FFN layer in FFN-Wider Transformers has a higher contribution ratio, we cannot assert this is the reason. This may merely be a correlation rather than a decisive factor in base capabilities.

Therefore, we designed an architecture that can directly intervene in the contribution ratios of the transformation and combination function, named Combination Adjustable Architecture (CAA).

## 5.1 Model Architecture

The new architecture, as shown in Figure5, differs from the FFN-Wider Transformer in two aspects: one is partially transferring the FFN into the MHA; the other is adding a direct pathway inside the MHA that bypasses the Inner-FFN, which we will introduce in detail below.

### 5.1.1 Inner-FFN and Outer-FFN

A wider FFN would increase the contribution ratio of transformation function to pre-trained language modeling. A natural idea: is it possible to devise a variant FFN serves only combination function?

We found the primary reason that the FFN does not serve only combination function is due to the residual connections. The representation transformed by the FFN does not necessarily have to go through the MHA combination process; the FFN can bypass the MHA via residual connections and directly transmit the information in the representation to subsequent layers.

Therefore, we divided an FFN into two parts with adjustable width ratios: one remains in its original position as a transformation function, known as the Outer-FFN, and the other is relocated within the MHA layer, transformed an Inner-FFN. This design ensures that the representation transformed by the Inner-FFN must undergo the MHA combination process before proceeding, initially achieving the goal of the Inner-FFN serving only for the combination function.

### 5.1.2 Direct Pathway in MHA

Although we considered the impact of residual connections and designed an Inner-FFN, the Inner-FFN still has a hidden pathway to bypass the combination function and directly transmit uncombined information to subsequent layers.

The attention mechanism involves weighted summation over values, and typically, the value at the current position also participates. This provides a possibility for the Inner-FFN to directly transmit information. Therefore, we designed a direct pathway in the MHA for current position computation to circumvent the Inner-FFN. Specifically, during the self-attention computation at the

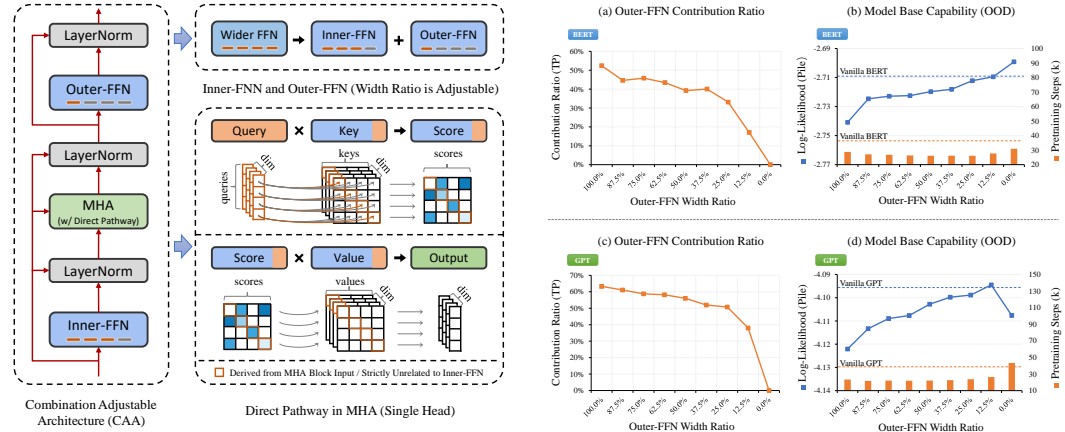

Figure 5: Overview of our proposed Combination Adjustable Architecture (CAA).

Figure 6: Outer-FFN contribution ratio and base capability under different width ratios.

current position, the query, key and value from the current position use the input representation that has not been processed by the Inner-FFN. Only the context representations from non-current positions are transformed by the Inner-FFN. This design largely eliminates the possibility of the Inner-FFN bypassing the combination function.

## 5.2 Width Adjustment Analysis

In this section, we control the width ratio of the Outer-FFN, reducing it gradually from 100% to 0%,confirming our hypothesis by examining the trends of contribution ratio and base capability.

### 5.2.1 Trend of Contribution Ratio

We conducted experiments on large-scale (H=768) BERT and GPT models. Initially, a series of models with different width ratios were pre-trained, ensuring their pre-training performance was aligned. Then, we calculated the contribution ratio of Outer-FFN using Token Prediction (TP) and plotted these trends, as shown in Figures 6(a) and 6(c).

It can be seen that the contribution ratio of Outer-FFN decreases as its width ratio is reduced. Conversely, the contribution ratio of the combination function increases with the reduction in the width ratio of Outer-FFN, demonstrating that controlling the width ratio indeed directly influences the contribution ratio of the combination function.

### 5.2.2 Trend of Base Capability

We tested the performance of each model above in out-of-distribution (OOD) language modeling and plotted the trend in Figures 6(b) and 6(d).

For BERT, the OOD performance improves as the Outer-FFN width decreases. The performance of the optimal model has already surpassed the vanilla BERT. Most models not only outperform the FFN-Wider BERT in terms of performance but also require fewer pre-training steps. For GPT, only the model without Outer-FFN showed anomalous results, other models also follow this pattern.

After the analyses, **a conclusion is drawn from the overall trend**: the model's base capabilities generally improve as the contribution ratio of combination function increases. It holds over a wide range of width ratios and proves our hypothesis.

However, this overall trend does not mean the contribution of transformation function should be reduced to zero. A minor contribution from transformation function might still be crucial, as indicated by the sole anomalous result in GPT. BERT was unaffected, which is related to bidirectional attention still being able to leak few transformation contribution. More details are in Appendix G.

| Model | Pile (22 parts) | GLUE (8 tasks) | SGLUE (8 tasks) | Others (6 tasks) |
|---|---|---|---|---|
| *(H=128, Pre-Training Performance≈ –2.61)* | | | | |
| Vanilla BERT | -3.211 | 78.45 | – | – |
| FFN-Wider BERT | -3.256 | 76.66 | – | – |
| FFN-Wider BERT w/ CEA | **-3.202** | **78.20** | – | – |
| - w/o Direct Pathway in MHA | -3.230 | 77.29 | – | – |
| *(H=768, Pre-Training Performance≈ –1.53)* | | | | |
| Vanilla BERT | -2.158 | 83.89 | 64.41 | 52.14 |
| FFN-Wider BERT | -2.196 | 82.60 | 62.82 | 51.61 |
| FFN-Wider BERT w/ CEA | **-2.149** | **83.86** | **64.22** | **53.12** |

Table 1: The results of various BERT models.

| Model | Pile (22 parts) | 0-Shot (9 tasks) | 1-Shot (9 tasks) |
|---|---|---|---|
| *(H=128, Pre-Training Performance≈ –4.06)* | | | |
| Vanilla GPT | -4.706 | – | – |
| FFN-Wider GPT | -4.728 | – | – |
| FFN-Wider GPT w/ CEA | **-4.685** | – | – |
| *(H=768, Pre-Training Performance≈ –3.19)* | | | |
| Vanilla GPT | -3.878 | 45.82 | 46.09 |
| FFN-Wider GPT | -3.911 | 44.60 | 44.84 |
| FFN-Wider GPT w/ CEA | **-3.882** | **45.52** | **45.84** |

Table 2: The results of various GPT models.

## 6 Combination Enhanced Architecture

In this section, we determine the Combination Enhanced Architecture (CEA) and pre-train more steps to verify whether it can reverse the decline in base capabilities of FFN-Wider Transformer.

### 6.1 Width Ratio Selection

We selected the optimal width ratio for Combination Enhanced Architecture (CEA). The Outer-FFN width ratio for FFN-Wider BERT w/ CEA is set to 0%. For FFN-Wider GPT w/ CEA, it is 12.5%.

### 6.2 Further Experiments

We conducted more pre-training steps on vanilla models, FFN-wider models, FFN-wider models w/ CEA, and aligned the pre-training performances.

For BERT, we tested fine-tuning performance on GLUE, SuperGLUE and other datasets. For GPT, we tested zero-shot and one-shot performance on multiple datasets. For both, we tested OOD language modeling on Pile. For small-scale models, we removed some experiments beyond their capabilities. The brief results are shown in Table 1 and 2, and the detailed results are in Appendix H.

The results show the FFN-wider models w/ CEA improve in base capabilities, not only surpassing the FFN-wider models in most aspects but also nearly reaching the level of vanilla models. In addition, the new architecture models w/o the direct pathway in MHA show a significant decline in base capabilities. These all confirm our explanation.

Additionally, we conducted similar experiments for other width ratios that can align pre-training performance and steps simultaneously, which can indisputably prove improvements come from architecture, as detailed in Appendix I.

## 7 From FFN-Wider Transformers to MoE Transformers

The Mixture of Experts (MoE) Transformer [18, 8] is a practical architecture that introduces a mix of experts, which enables expanding the capacity with lower computation expense.

However, base capability decline also be observed in MoE Transformers. [8] found MoE models perform lower on SuperGLUE compared to vanilla models when achieving same pre-training level. Similar issue can also be analyzed from the results of [1].

We found the MoE layer can be seen as an enhanced version of the FFN layer. Therefore, we believe the previous explanations could also apply to MoE Transformers. Consequently, we directly transplant our CEA to MoE Transformers, resulting in a new MoE architecture.

We selected a 1.3B GPT model as the backbone model, which incorporates Rotary Embedding [30] and RMSNorm [37] on the GPT3 [3]. Then, we added a MoE layer with 64 experts (top-1 activation) before the FFN layer, resulting in a 14B parameter MoE baseline model. Finally, our improved version with CEA involves transforming the MoE layer into an Inner-MoE layer within the MHA,

| Metric | # Shot | Vanilla GPT 1.3B | Vanilla MoE 14B | MoE 14B w/ CEA |
|---|---|---|---|---|
| # Total Params | N/A | 1.3B | 14B | 14B |
| # Activated Params | N/A | 1.3B | 1.5B | 1.5B |
| # Total Experts | N/A | - | 64 | 64 |
| # Activated Experts | N/A | - | 1 | 1 |
| # Training Tokens | N/A | 100B | 100B | 100B |
| SlimPajama-CC&C4 (Loss)† | N/A | 2.382 | 2.315 | **2.303** |
| SlimPajama-Arxiv (Loss) | N/A | 2.353 | 2.320 | **2.239** |
| SlimPajama-Book (Loss) | N/A | 2.796 | 2.761 | **2.585** |
| SlimPajama-Github (Loss) | N/A | 1.989 | 1.989 | **1.840** |
| SlimPajama-Stack (Loss) | N/A | 2.653 | 2.588 | **2.465** |
| SlimPajama-Wiki (Loss) | N/A | 3.137 | 3.155 | **2.964** |
| LAMBADA (PPL) | 0-shot | 23.7 | 24.8 | **19.3** |
| LAMBADA (Acc.) | 0-shot | 36.6 | 36.4 | **39.7** |
| MMLU (Acc.) | 5-shot | 30.8 | 30.6 | **31.4** |
| OpenBookQA (Acc.) | 5-shot | 36.6 | 35.7 | **37.2** |
| ARC Easy (Acc.) | 5-shot | 57.5 | 56.8 | **59.2** |
| ARC Challenge (Acc.) | 5-shot | 31.6 | 30.0 | **32.3** |
| BoolQ (Acc.) | 5-shot | **62.2** | **62.2** | 62.1 |
| RACE Middle (Acc.) | 5-shot | 42.3 | 44.4 | **45.5** |
| RACE High (Acc.) | 5-shot | 36.1 | 36.0 | **36.6** |
| SIQA (Acc.) | 5-shot | 41.0 | 41.7 | **43.7** |
| SCIQ (Acc.) | 5-shot | 78.1 | 72.7 | **84.1** |
| HellaSwag (Acc.) | 5-shot | 43.7 | 45.8 | **47.5** |
| COPA (Acc.) | 5-shot | 69.2 | 69.2 | **70.4** |
| PIQA (Acc.) | 5-shot | 70.4 | 70.6 | **71.5** |
| StoryCloze (Acc.) | 5-shot | 67.7 | 68.0 | **69.5** |
| WinoGrande (Acc.) | 5-shot | 52.7 | 53.1 | **53.4** |
| Winograd (Acc.) | 5-shot | **67.2** | 66.7 | 66.8 |

Table 3: The results of Vanilla GPT 1.3B, Vanilla MoE 14B and MoE 14B w/ CEA. † indicates the test set drawn from the same distribution as the pre-training data.

while retaining the original FFN layer as the Outer-FFN layer. To accommodate FlashAttention-2 [6], we simplified the direct pathway. Instead of preventing transformation leakage by replacing the key and value at current position, we chose to directly mask the current position to achieve the same effect. The specifications, pre-training procedures and few-shot learning procedures are detailed in Appendix C and E.

We conducted pre-training on CC and C4 subsets within SlimPajama [29], training all models from scratch with 100B tokens. Due to the de-duplication across subsets achieved by SlimPajama, we used the remaining subsets as out-of-distribution sets. We carried out comprehensive out-of-distribution language modeling and few-shot learning tests, with the results presented in Table 3.

From the results, it is evident that our method significantly enhances the base capabilities of the MoE model. This fully demonstrates the effectiveness of our analysis and improvement methods.

# 8   Limitations

This work mainly analyzes the models that use language modeling as the pre-training objective, lacking experiments on models with other pre-training objectives. Hence, the conclusions are limited to pre-trained language models. Therefore, the current applicability of our findings is relatively narrow, and we consider conducting more experiments in the future.

# 9 Conclusion

This work explores how architecture affects the base capabilities of pre-trained language models. FFN-Wider Transformer is our research object and we try to explain and reverse the decline in base capabilities caused by its architecture. We found the contribution ratio of combination function is a key factor, while FFN-Wider Transformer reduces it, leading to a decline in base capabilities. We solved it by proposing CEA. In addition, we extended our conclusion to MoE Transformers, proving our work can offer guidance for architecture improvement.

## Acknowledgments

This work was supported by the Fundamental Research Funds for the Central Universities (project number: 2022FRFK060002), the National Key RD Program of China via grant 2021YFF0901602 and the National Natural Science Foundation of China (NSFC) via grant 62176078.

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

# A    Other Evaluation Schemes

Aside from the pre-training performance alignment scheme we adopted, there are three other schemes. However, none of these are suitable for our work.

**Pre-training steps alignment**    Differences in model's base capabilities might be due to differences in model capacity, not solely from architecture changes. Especially in models with different architectures and parameter scales, models with more parameters have a capacity advantage, rapidly reducing pre-training loss to a low level, showing good metrics on various tasks, and appearing to have strong base capabilities. But this is more due to large model capacity, and not much related to architecture inductive biases.

**Parameter numbers alignment**    It imposes restrictions on the model that make it inapplicable to any two different architecture models. More critically, this scheme has clear counterexamples, as it is not suitable for ALBERT models [17]. Under the same parameter numbers, ALBERT models exhibit strong base capabilities, but this is due to the large computational load, not much related to architecture inductive biases.

**Computational load alignment**    It imposes restrictions on the model that make it inapplicable to any two different architecture models. More critically, this scheme also has clear counterexamples, as it is not suitable for MoE models [18, 8]. Under the same computational load, MoE models demonstrate strong base capabilities, but this is due to large parameter numbers and model capacity, not much related to architecture inductive biases.

Although these schemes have shortcomings, they also have reasonable aspects. To indisputably prove base capability improvements come from our architecture changes, we also conducted experiments that align pre-training performance, pre-training steps, parameter numbers and computational load simultaneously, as detailed in Appendix I.

In addition, the experiment with MoE transformers also adopts a stricter setting similar to the above. In the comparative experiments between the vanilla MoE model and the improved MoE model, not only is pre-training performance aligned, but pre-training steps, parameter numbers, and computational load are also kept consistent.

# B    Evaluation Tasks

## B.1    FFN-Wider Transformers

All experiments with FFN-Wider transformers follow the unified settings described here:

**The pre-training corpus** is consistent with that of BERT [7], namely Wikipedia and BooksCorpus. We partition a portion of the corpus to serve as an in-distribution development set, and use it to align pre-training performance.

We conduct experiments on four specifications: BERT (H=128), BERT (H=768), GPT (H=128) and GPT (H=768). The specifications and pre-training procedures are detailed in Appendix C.

**For out-of-distribution language modeling capability**, we evaluate on the development set of Pile dataset [9].

**For transfer learning capability**, we only evaluate BERT models on GLUE [34], SuperGLUE [33], HellaSwag [36], PIQA [2], OpenBookQA [20], ARC Easy & Challenge [5] and WinoGrande [27]. The experimental settings are detailed in Appendix D.

**For few-shot learning capability**, we only evaluate GPT models. Limited by the maximum sequence length (128) of our pre-trained models, we conduct 0-shot and 1-shot experiments on HellaSwag, PIQA, OpenBookQA, ARC Easy & Challenge, WinoGrande, Winograd [19], COPA [26] and StoryCloze [21]. The experimental settings are detailed in Appendix E.

## B.2    MoE Transformers

All experiments with MoE transformers follow the unified settings described here:

**The pre-training corpus** is the CC and C4 subsets within the SlimPajama dataset [29]. The specifications and pre-training procedures are detailed in Appendix C.

**For out-of-distribution language modeling capability**, due to the de-duplication across subsets achieved by SlimPajama [29], we used the remaining subsets (Arxiv, Book, Github, Stack and Wiki) as out-of-distribution test sets.

**For few-shot learning capability**, we conduct 0-shot experiments on LAMBADA [23], and conduct 5-shot experiments on MMLU [10], OpenBookQA, ARC Easy & Challenge, BoolQ [4], RACE Middle & High [16], SIQA [28], SCIQ [35], HellaSwag, COPA, PIQA, StoryCloze, WinoGrande and Winograd. The experimental settings are detailed in Appendix E.

## C   Model Specifications and Pre-training Procedures

### C.1   FFN-Wider Transformers

All experiments were conducted in English only. We concurrently pre-trained both BERT and GPT models, where the architecture design of the BERT model adheres to the work of [7], and the GPT model structure follows that established by [24]. Both models employ a post layer normalization scheme.

We trained small and large scales of both BERT and GPT. The small-scale models have a hidden dimension of 128, 12 layers, and 2 attention heads; the large-scale models possess a hidden dimension of 768, 12 layers, and 12 attention heads. In vanilla BERT and GPT, the FFN intermediate dimension is 4 times the hidden dimension, while the ratio of FFN-Wider models is 32 times. The small-scale vanilla models have 6.3M Parameters and the small-scale other models have 17.3M Parameters. The large-scale vanilla models have 110M Parameters and the large-scale other models have 506M Parameters. All models have a maximum sequence length of 128 and utilize the BERT vocabulary released by [7], comprising 30,522 tokens. We used PyTorch[2] and transformers[3] libraries.

The BERT model employs a masked language modeling task for pre-training, masking 15% of tokens in the sequence—80% are replaced with [MASK], 10% with random tokens, and 10% remain unchanged. Differing from [7], we removed the next sentence prediction task, using long and continuous text for pre-training inputs and applying different masking schemes to the same input sequence in various epochs. The GPT model is pre-trained using a language modeling task without additional special settings.

The maximum epoch set for all model pre-training is 40, but in practice, it was not reached; mid-training checkpoints were used for alignment and experimentation. All models used the Adam optimizer [15] for pre-training, with a learning rate of 1e-4, $\beta_1 = 0.9$, $\beta_2 = 0.999$, L2 weight of 0.01, a warm-up over the first 10,000 steps, followed by linear decay. The small-scale models were pre-trained with a batch size of 512 on four Nvidia Tesla V100s, and total GPU days are approximately 55 days; the large-scale models with a batch size of 1024 on four Nvidia Tesla A100s, and total GPU days are approximately 87 days.

To fairly compare the base capabilities of different architecture models, all models were pre-trained from scratch. However, as mentioned earlier, due to limited computational resources, our pre-training steps generally fell short of those in the original papers [7, 24, 25, 3], which may result in discrepancies in downstream task performance compared to the original research.

### C.2   MoE Transformers

All experiments were conducted in English only.

We selected the GPT model as the backbone model (Vanilla GPT 1.3B), which incorporates Rotary Embedding [30] and RMSNorm [37] on the GPT3 [3]. Then, we added a MoE layer before the FFN layer, resulting in our MoE baseline model (Vanilla MoE 14B). Finally, our improved version with CEA (MoE 14B w/ CEA) involves transforming the MoE layer into an Inner-MoE layer within the MHA, while retaining the original FFN layer as the Outer-FFN layer. To accommodate

---

[2]https://pytorch.org/
[3]https://github.com/huggingface/transformers

FlashAttention-2 [6], we simplified the direct pathway. Instead of preventing transformation leakage by replacing the key and value at current position, we chose to directly mask the current position to achieve the same effect.

The Vanilla GPT 1.3B has a hidden dimension of 4,096, 6 layers and 32 attention heads; the Vanilla MoE 14B has a hidden dimension of 4,096, 6 layers, 32 attention heads and 6 MoE layers with 64 experts (top-1 activation, intermediate dimension is 4,096) before the corresponding FFN layers; the MoE 14B w/ CEA is similar to the Vanilla MoE 14B, just the position of the parameters has changed. All models have a maximum sequence length of 2,048 and utilize the Mixtral vocabulary released by [13], comprising 32,000 tokens. We used PyTorch, transformers and fastmoe[4] libraries.

These models are pre-trained using a language modeling task without additional special settings. We conducted pre-training on CC and C4 subsets within SlimPajama [29], training all models from scratch with 100B tokens. Due to the de-duplication across subsets achieved by SlimPajama, we used the remaining subsets as out-of-distribution sets. All models used the Adam optimizer [15] for pre-training, with a learning rate of 5e-4, $\beta_1 = 0.9$, $\beta_2 = 0.95$, L2 weight of 0.01, a warm-up over the first 2,500 steps, followed by linear decay. All models were pre-trained with a batch size of 256 on 8 Nvidia Tesla A100 80G cards, and total GPU days are approximately 280 days.

## D    Fine-tuning Settings

We conducted fine-tuning experiments on BERT models across various datasets. For the small-scale (H=128) models, we only conducted experiments on GLUE, while for the large-scale (H=768) models, we experimented with all datasets.

The maximum number of training epochs during fine-tuning was 10, with a batch size of 32. The optimizer was Adam [15], with a warmup ratio of 0.06, a linearly decaying learning rate, and a weight decay of 0.01. We reported the average performance of multiple runs.

For the small-scale (H=128) models, we observed models with a wider FFN might overfit when fully fine-tuned with all parameters. Thus, we performed both full parameter fine-tuning and efficient fine-tuning based on adapters [12] for all models, choosing the better result of the two for reporting. For full parameter fine-tuning, the learning rates were {1e-5, 2e-5, 5e-5}; for efficient fine-tuning, the adapter size was 128, with learning rates of {1e-4, 2e-4, 3e-4}. For the large-scale (H=768) models, we directly conducted full parameter fine-tuning for all models, with learning rates of {1e-5, 2e-5, 5e-5}.

## E    Few-shot Learning Settings

In our study on GPT and MoE models, we conducted few-shot learning experiments on multiple datasets. Since small-scale (H=128) models have limited capabilities and struggle with few-shot learning, we focused our experiments only on large-scale (H=768) models for 0-shot and 1-shot learning and MoE models for 0-shot and 5-shot learning.

We followed the method of [3], which involves transforming the classification into comparisons of probability magnitudes.

Specifically, the unified format of these datasets involves selecting one option from multiple choices, given a context. We concatenated the context with different options and compared their respective generation probabilities. This comparison could be based on either the probability of the option alone or the entire sequence text, with a choice of normalizing the probabilities by length or not. [3] mentioned a method of normalizing the probability of each option unconditionally, which we also included in our options. We then experimented with all possible combinations of these choices for each dataset, determined the best combination for each, and reported the results of all models under these optimal conditions.

For 1-shot experiment, we randomly selected one demonstration each time, repeated the experiment 10 times, and reported the average results. For 5-shot experiment, we randomly selected five demonstrations each time (for MMLU, randomly shuffling the order of demonstrations), repeated the experiment 5 times, and reported the average results.

---

[4]https://github.com/laekov/fastmoe

## F  Mutual Information Estimate

The formal definition of mutual information for two discrete random variables is as follows:

$$I(X;Y) = \sum_{y \in \mathcal{Y}} \sum_{x \in \mathcal{X}} P_{(X,Y)}(x,y) \log\left(\frac{P_{(X,Y)}(x,y)}{P_X(x)P_Y(y)}\right) \tag{1}$$

where $X$ and $Y$ are discrete random variables, $P_{(X,Y)}$ is the joint probability mass function of $X$ and $Y$, and $P_X$ and $P_Y$ are the marginal probability mass functions of $X$ and $Y$ respectively.

For estimating the mutual information between intermediate representations and target tokens, we adopted the method proposed by [32], which primarily involves transforming the representations into discrete variables through clustering and then calculating mutual information.

We first randomly sampled 6.94 million input tokens and their corresponding output tokens from a pre-trained development set and collected all intermediate representations of the model at these locations. Then, for the GPT model, we performed clustering directly at each layer and calculated mutual information using the labels obtained from clustering with the output tokens. For the BERT model, we also conducted clustering at each layer; however, since masked language modeling only considers masked positions as the actual output, mutual information was calculated solely at these masked positions. Clustering was done using the mini-batch k-means algorithm, initialized with the k-means++ method, with a batch size of 1024 and a class number set to 2000.

## G  Additional Transformation Function Contribution in BERT

While it appears that BERT can reduce the actual contribution ratio of transformation function to zero, in reality, some contributions of transformation function remain uncovered by our method.

For BERT, although we have eliminated most of the pathways through which the Inner-FFN directly leaks independent transformation information to subsequent layers, there is still one way this can happen: due to the bidirectional nature of attention, in the MHA of the current block, the entire sequence context has already been infiltrated by information about the current position at lower levels. Therefore, the Inner-FFN can independently enhance this part of the information, which can then leak out again through the weighted summation over values. In contrast, the GPT models employ unidirectional attention, so the context definitely does not contain any information about the current position, hence there is truly no independent transformation without the Outer-FFN.

Therefore, while we are fairly certain that reducing the contribution ratio of independent transformation function is a general trend that can enhance base capabilities, completely eliminating the contribution of it could be harmful, as indicated by the trend in GPT; the absence of this phenomenon in BERT's trend is merely due to the likelihood that BERT still retains a small part of independent transformation in the Inner-FFN.

## H  Detailed Results (Main Setting: Pre-training Performance Alignment)

In Section 6.2, we present the brief experimental results in Table 1 and 2, and here we provide the corresponding detailed results.

For BERT, out-of-distribution language modeling results on Pile are shown in Table 6, fine-tuning results on GLUE Benchmark are shown in Table 7, fine-tuning results on SuperGLUE Benchmark are shown in Table 8, fine-tuning results on multiple other tasks are shown in Table 9.

For GPT, out-of-distribution language modeling results on Pile are shown in Table 10, zero-shot results and one-shot results on multiple datasets are shown in Table 11.

## I  Detailed Results (Extra Setting: Pre-training Performance Alignment & Pre-training Steps Alignment)

The main experiments in this work are based on the pre-training performance alignment scheme. Although we have introduced the rationale behind this scheme, the difference in pre-training steps

| Model | Pile (22 parts) | GLUE (8 tasks) | SGLUE (8 tasks) | Others (6 tasks) |
|---|---|---|---|---|
| *(H=128, Pre-Training Performance≈ −2.61, Pre-Training Steps= 155.2k)* | | | | |
| FFN-Wider BERT | -3.256 | 76.66 | – | – |
| FFN-Wider BERT w/ CAA (12.5%) | **-3.203** | **78.08** | – | – |
| *(H=768, Pre-Training Performance≈ −1.53, Pre-Training Steps= 164.9k)* | | | | |
| FFN-Wider BERT | -2.196 | 82.60 | 62.82 | 51.61 |
| FFN-Wider BERT w/ CAA (12.5%) | **-2.153** | **83.48** | **63.79** | **52.45** |

Table 4: The results of BERT (Pre-training Steps Alignment & Pre-training Performance Alignment).

| Model | Pile (22 parts) | 0-Shot (9 tasks) | 1-Shot (9 tasks) |
|---|---|---|---|
| *(H=128, Pre-Training Performance≈ −4.06, Pre-Training Steps= 79.2k)* | | | |
| FFN-Wider GPT | -4.728 | – | – |
| FFN-Wider GPT w/ CAA (37.5%) | **-4.706** | – | – |
| *(H=768, Pre-Training Performance≈ −3.19, Pre-Training Steps= 50.9k)* | | | |
| FFN-Wider GPT | -3.911 | 44.60 | 44.84 |
| FFN-Wider GPT w/ CAA (37.5%) | **-3.893** | **45.27** | **45.82** |

Table 5: The results of GPT (Pre-training Steps Alignment & Pre-training Performance Alignment).

between original FFN-Wider models and new architecture models may still raise question: is the base capability improvement of new architecture models really due to architecture changes?

To answer this question, we chose the Outer-FFN width ratios that allow CAA to align with FFN-Wider Transformer for both pre-training performance and pre-training steps, and conducted experiments similar to those in Section 6.2. Moreover, the parameter numbers and computational load of CAA are also aligned with the FFN-Wider Transformer. Therefore, we achieved alignment in four aspects, and under this setting, the improvements of base capabilities can be incontrovertibly attributed to our architectural changes.

The Outer-FFN width ratio for FFN-Wider BERT w/ CAA is set to 12.5%, and for FFN-Wider GPT w/ CAA, it is set to 37.5%.

The brief results are shown in Table 4 and 5, and the corresponding detailed results are as follows:

For BERT, out-of-distribution language modeling results on Pile are shown in Table 12, fine-tuning results on GLUE Benchmark are shown in Table 13, fine-tuning results on SuperGLUE Benchmark are shown in Table 14, fine-tuning results on multiple other tasks are shown in Table 15.

For GPT, out-of-distribution language modeling results on Pile are shown in Table 16, zero-shot results and one-shot results on multiple datasets are shown in Table 17.

The experimental results show that the base capabilities of new architecture models have still significantly improved, confirming the positive impact of our architecture modifications on base capabilities.

| | H=128 | | | | H=768 | | |
|---|---|---|---|---|---|---|---|
| | Vanilla BERT | FFN-Wider BERT | FFN-Wider BERT w/ CEA | - w/o Direct Pathway in MHA | Vanilla BERT | FFN-Wider BERT | FFN-Wider BERT w/ CEA |
| Validation | -2.617 | -2.613 | -2.614 | -2.611 | -1.532 | -1.536 | -1.538 |
| ArXiv | -3.288 | -3.405 | -3.301 | -3.397 | -2.315 | -2.369 | -2.242 |
| BookCorpus2 | -2.636 | -2.623 | -2.629 | -2.621 | -1.572 | -1.583 | -1.569 |
| Books3 | -3.018 | -3.031 | -3.014 | -3.012 | -1.879 | -1.904 | -1.886 |
| DM Mathematics | -2.780 | -2.865 | -2.730 | -2.878 | -2.149 | -2.330 | -2.158 |
| Enron Emails | -3.160 | -3.181 | -3.134 | -3.152 | -2.100 | -2.135 | -2.105 |
| EuroParl | -4.804 | -4.862 | -4.836 | -4.851 | -3.492 | -3.474 | -3.420 |
| FreeLaw | -3.099 | -3.126 | -3.087 | -3.105 | -1.935 | -1.952 | -1.941 |
| Github | -3.310 | -3.391 | -3.260 | -3.321 | -2.371 | -2.446 | -2.345 |
| Gutenberg (PG-19) | -3.172 | -3.181 | -3.179 | -3.174 | -2.090 | -2.127 | -2.117 |
| HackerNews | -3.223 | -3.273 | -3.203 | -3.211 | -2.212 | -2.247 | -2.219 |
| NIH ExPorter | -2.984 | -3.036 | -2.991 | -3.004 | -1.781 | -1.798 | -1.778 |
| OpenSubtitles | -2.080 | -2.091 | -2.066 | -2.063 | -1.391 | -1.413 | -1.406 |
| OpenWebText2 | -3.313 | -3.336 | -3.314 | -3.311 | -2.128 | -2.149 | -2.126 |
| PhilPapers | -4.171 | -4.221 | -4.188 | -4.201 | -2.872 | -2.853 | -2.823 |
| Pile-CC | -3.229 | -3.248 | -3.222 | -3.219 | -2.096 | -2.127 | -2.106 |
| PubMed Abstracts | -2.884 | -2.952 | -2.880 | -2.911 | -1.719 | -1.746 | -1.710 |
| PubMed Central | -2.949 | -3.007 | -2.941 | -2.977 | -1.942 | -1.989 | -1.933 |
| Stack Exchange | -3.379 | -3.440 | -3.347 | -3.387 | -2.389 | -2.436 | -2.374 |
| Ubuntu IRC | -3.982 | -4.035 | -3.945 | -4.033 | -3.008 | -3.105 | -3.032 |
| USPTO Backgrounds | -2.824 | -2.878 | -2.812 | -2.833 | -1.774 | -1.823 | -1.785 |
| Wikipedia (en) | -2.790 | -2.813 | -2.780 | -2.791 | -1.663 | -1.674 | -1.643 |
| YoutubeSubtitles | -3.566 | -3.630 | -3.583 | -3.608 | -2.594 | -2.642 | -2.563 |
| Average | -3.211 | -3.256 | -3.202 | -3.230 | -2.158 | -2.196 | -2.149 |

Table 6: Out-of-distribution language modeling results on the development set of Pile of various BERT models (Pre-training Performance Alignment).

| Model | CoLA | MRPC | SST-2 | STS-B | RTE | MNLI | QNLI | QQP | Avg. |
|---|---|---|---|---|---|---|---|---|---|
| *(H=128, Pre-Training Performance≈ –2.61)* | | | | | | | | | |
| Vanilla BERT | 35.46 | 83.01 | 88.18 | 88.30 | 83.89 | 83.57 | 65.59 | 77.13 | 77.59 | 84.43 | 89.09 | 85.15 | 78.45 |
| FFN-Wider BERT | **34.95** | 79.94 | 85.79 | 87.84 | 81.85 | 81.76 | 59.20 | 76.10 | 76.22 | 83.43 | 88.53 | 84.35 | 76.66 |
| FFN-Wider BERT w/ CEA | 33.70 | **83.17** | **88.22** | 88.19 | 82.96 | 82.78 | 64.26 | **77.72** | **78.07** | **84.65** | **89.15** | **85.50** | **78.20** |
| - w/o Direct Pathway in MHA | 30.14 | 81.82 | 87.15 | 87.73 | 82.89 | 82.61 | 62.76 | 77.36 | 77.65 | 83.27 | 88.99 | 85.12 | 77.29 |
| *(H=768, Pre-Training Performance≈ –1.53)* | | | | | | | | | |
| Vanilla BERT | 62.10 | 87.33 | 90.70 | 92.43 | 87.97 | 87.74 | 62.04 | 83.43 | 83.78 | 90.06 | 91.08 | 88.02 | 83.89 |
| FFN-Wider BERT | 61.66 | 85.42 | 89.73 | 91.86 | 86.35 | 86.17 | 60.79 | 81.87 | 82.24 | 88.69 | 89.98 | 86.49 | 82.60 |
| FFN-Wider BERT w/ CEA | **62.58** | **87.99** | **91.45** | 92.32 | 87.30 | 87.13 | 62.69 | 83.01 | 82.81 | 89.86 | 91.10 | 88.08 | 83.86 |

Table 7: Fine-tuning results on the development set of GLUE Benchmark of various BERT models (Pre-training Performance Alignment).

| Model | BoolQ | CB | COPA | MultiRC | WiC | ReCoRD | WSC | RTE | Avg. |
|---|---|---|---|---|---|---|---|---|---|
| *(H=768, Pre-Training Performance≈ –1.53)* | | | | | | | | | |
| Vanilla BERT | 75.44 | 86.01 | 83.88 | 64.83 | 61.91 | 16.24 | 66.77 | 63.70 | 62.82 | 64.90 | 62.04 | 64.41 |
| FFN-Wider BERT | 73.70 | 83.04 | 79.68 | 64.71 | 60.90 | 14.57 | 64.62 | 62.54 | **61.73** | 64.78 | 60.79 | 62.82 |
| FFN-Wider BERT w/ CEA | **74.76** | **85.36** | 82.65 | **66.50** | 61.64 | 16.04 | **66.59** | 62.61 | 61.43 | **66.11** | 62.69 | **64.22** |

Table 8: Fine-tuning results on the development set of SuperGLUE Benchmark of various BERT models (Pre-training Performance Alignment).

| Model | HellaSwag | PIQA | WinoGrande | OpenBookQA | ARC Easy | ARC Chal. | Avg. |
|---|---|---|---|---|---|---|---|
| *(H=768, Pre-Training Performance≈ –1.53)* | | | | | | | |
| Vanilla BERT | 40.79 | 67.46 | 58.06 | 56.38 | 53.47 | 36.65 | 52.14 |
| FFN-Wider BERT | 38.95 | 67.00 | 55.09 | 57.78 | 51.74 | 39.07 | 51.61 |
| FFN-Wider BERT w/ CEA | 40.16 | **68.14** | **58.87** | **58.71** | 52.50 | **40.34** | **53.12** |

Table 9: Fine-tuning results on multiple other tasks of various BERT models (Pre-training Performance Alignment).

|  | H=128 | | | H=768 | | |
|---|---|---|---|---|---|---|
|  | Vanilla GPT | FFN-Wider GPT | FFN-Wider GPT w/ CEA | Vanilla GPT | FFN-Wider GPT | FFN-Wider GPT w/ CEA |
| Validation | -4.061 | -4.067 | -4.066 | -3.197 | -3.191 | -3.192 |
| ArXiv | -4.855 | -4.927 | -4.893 | -3.981 | -4.061 | -3.980 |
| BookCorpus2 | -4.067 | -4.055 | -4.061 | -3.309 | -3.315 | -3.301 |
| Books3 | -4.499 | -4.495 | -4.486 | -3.700 | -3.715 | -3.697 |
| DM Mathematics | -4.110 | -4.162 | -3.999 | -3.542 | -3.597 | -3.526 |
| Enron Emails | -4.619 | -4.636 | -4.583 | -3.846 | -3.877 | -3.831 |
| EuroParl | -6.006 | -6.026 | -5.987 | -4.853 | -4.855 | -4.888 |
| FreeLaw | -4.625 | -4.625 | -4.599 | -3.711 | -3.726 | -3.703 |
| Github | -4.979 | -5.030 | -4.899 | -4.135 | -4.189 | -4.130 |
| Gutenberg (PG-19) | -4.618 | -4.610 | -4.600 | -3.909 | -3.930 | -3.910 |
| HackerNews | -4.743 | -4.738 | -4.711 | -4.049 | -4.061 | -4.057 |
| NIH ExPorter | -4.612 | -4.617 | -4.610 | -3.632 | -3.657 | -3.636 |
| OpenSubtitles | -3.338 | -3.345 | -3.316 | -2.875 | -2.886 | -2.874 |
| OpenWebText2 | -4.810 | -4.817 | -4.804 | -3.958 | -3.975 | -3.962 |
| PhilPapers | -5.558 | -5.580 | -5.547 | -4.499 | -4.522 | -4.525 |
| Pile-CC | -4.754 | -4.758 | -4.747 | -3.962 | -3.979 | -3.962 |
| PubMed Abstracts | -4.575 | -4.591 | -4.562 | -3.569 | -3.593 | -3.570 |
| PubMed Central | -4.583 | -4.597 | -4.521 | -3.760 | -3.797 | -3.766 |
| Stack Exchange | -4.989 | -5.031 | -4.964 | -4.182 | -4.233 | -4.187 |
| Ubuntu IRC | -5.649 | -5.729 | -5.660 | -4.857 | -4.989 | -4.897 |
| USPTO Backgrounds | -4.460 | -4.503 | -4.453 | -3.636 | -3.690 | -3.642 |
| Wikipedia (en) | -4.302 | -4.312 | -4.295 | -3.345 | -3.359 | -3.339 |
| YoutubeSubtitles | -4.782 | -4.822 | -4.775 | -3.999 | -4.036 | -4.013 |
| Average | -4.706 | -4.728 | -4.685 | -3.878 | -3.911 | -3.882 |

Table 10: Out-of-distribution language modeling results on the development set of Pile of various GPT models (Pre-training Performance Alignment).

| Model | HellaSwag | PIQA | WinoGrande | COPA | OpenBookQA | ARC Easy | ARC Chal. | StoryCloze | Winograd | Avg. |
|---|---|---|---|---|---|---|---|---|---|---|
| *(Random Performance)* | 25.00 | 50.00 | 50.00 | 50.00 | 25.00 | 25.00 | 25.00 | 50.00 | 50.00 | 38.89 |
| *(H=768, 0-Shot, Pre-Training Performance≈–3.19)* | | | | | | | | | | |
| Vanilla GPT | 26.67 | 57.28 | 52.80 | 61.00 | 30.40 | 44.80 | 25.85 | 55.69 | 57.90 | 45.82 |
| FFN-Wider GPT | 26.79 | **57.77** | 51.62 | 57.00 | **29.20** | 43.56 | 22.45 | 56.49 | 56.49 | 44.60 |
| FFN-Wider GPT w/ CEA | **27.12** | 57.28 | **52.25** | **60.00** | 28.00 | **45.50** | **24.49** | **57.51** | **57.54** | **45.52** |
| *(H=768, 1-Shot, Pre-Training Performance≈–3.19)* | | | | | | | | | | |
| Vanilla GPT | 27.89 | 56.89 | 50.94 | 63.60 | 29.36 | 41.82 | 28.06 | 56.32 | 59.97 | 46.09 |
| FFN-Wider GPT | 27.54 | 57.18 | 50.33 | **63.10** | 28.36 | 40.72 | 25.20 | 56.24 | 54.88 | 44.84 |
| FFN-Wider GPT w/ CEA | **28.21** | **57.22** | **52.25** | 61.50 | **28.68** | **42.31** | 27.38 | **57.37** | **57.65** | **45.84** |

Table 11: Zero-shot and one-shot results on multiple datasets of various GPT models (Pre-training Performance Alignment).

| | H=128 | | H=768 | |
|---|---|---|---|---|
| | FFN-Wider BERT | FFN-Wider BERT w/ CAA (12.5%) | FFN-Wider BERT | FFN-Wider BERT w/ CAA (12.5%) |
| Validation | -2.613 | -2.610 | -1.536 | -1.531 |
| ArXiv | -3.405 | -3.305 | -2.369 | -2.220 |
| BookCorpus2 | -2.623 | -2.627 | -1.583 | -1.578 |
| Books3 | -3.031 | -3.011 | -1.904 | -1.887 |
| DM Mathematics | -2.865 | -2.722 | -2.330 | -2.158 |
| Enron Emails | -3.181 | -3.144 | -2.135 | -2.095 |
| EuroParl | -4.862 | -4.853 | -3.474 | -3.472 |
| FreeLaw | -3.126 | -3.087 | -1.952 | -1.938 |
| Github | -3.391 | -3.295 | -2.446 | -2.358 |
| Gutenberg (PG-19) | -3.181 | -3.165 | -2.127 | -2.107 |
| HackerNews | -3.273 | -3.213 | -2.247 | -2.209 |
| NIH ExPorter | -3.036 | -3.005 | -1.798 | -1.787 |
| OpenSubtitles | -2.091 | -2.060 | -1.413 | -1.407 |
| OpenWebText2 | -3.336 | -3.314 | -2.149 | -2.126 |
| PhilPapers | -4.221 | -4.195 | -2.853 | -2.860 |
| Pile-CC | -3.248 | -3.222 | -2.127 | -2.097 |
| PubMed Abstracts | -2.952 | -2.908 | -1.746 | -1.727 |
| PubMed Central | -3.007 | -2.939 | -1.989 | -1.931 |
| Stack Exchange | -3.440 | -3.336 | -2.436 | -2.364 |
| Ubuntu IRC | -4.035 | -3.882 | -3.105 | -3.012 |
| USPTO Backgrounds | -2.878 | -2.811 | -1.823 | -1.783 |
| Wikipedia (en) | -2.813 | -2.781 | -1.674 | -1.662 |
| YoutubeSubtitles | -3.630 | -3.594 | -2.642 | -2.599 |
| Average | -3.256 | -3.203 | -2.196 | -2.153 |

Table 12: Out-of-distribution language modeling results on the development set of Pile of various BERT models (Pre-training Steps Alignment & Pre-training Performance Alignment).

| Model | CoLA | MRPC | SST-2 | STS-B | RTE | MNLI | QNLI | QQP | | Avg. |
|---|---|---|---|---|---|---|---|---|---|---|
| *(H=128, Pre-Training Performance≈ –2.61, Pre-Training Steps= 155.2k)* | | | | | | | | | | |
| FFN-Wider BERT | 34.95 | 79.94 | 85.79 | **87.84** | 81.85 | 81.76 | 59.20 | 76.10 | 76.22 | 83.43 | 88.53 | 84.35 | 76.66 |
| FFN-Wider BERT w/ CAA (12.5%) | **35.77** | **83.44** | **88.48** | 87.20 | **84.31** | **84.07** | **59.51** | **77.33** | **78.05** | **84.14** | **89.11** | **85.50** | **78.08** |
| *(H=768, Pre-Training Performance≈ –1.53, Pre-Training Steps= 164.9k)* | | | | | | | | | | |
| FFN-Wider BERT | 61.66 | 85.42 | 89.73 | **91.86** | 86.35 | 86.17 | 60.79 | 81.87 | 82.24 | 88.69 | 89.98 | 86.49 | 82.60 |
| FFN-Wider BERT w/ CAA (12.5%) | **61.72** | **87.13** | **90.88** | 91.06 | **87.71** | **87.58** | **61.37** | **83.38** | **83.12** | **89.29** | **90.83** | **87.71** | **83.48** |

Table 13: Fine-tuning results on the development set of GLUE Benchmark of various BERT models (Pre-training Steps Alignment & Pre-training Performance Alignment).

| Model | BoolQ | CB | COPA | MultiRC | WiC | ReCoRD | WSC | RTE | Avg. |
|---|---|---|---|---|---|---|---|---|---|
| *(H=768, Pre-Training Performance≈ –1.53, Pre-Training Steps= 164.9k)* | | | | | | | | | |
| FFN-Wider BERT | 73.70 | 83.04 | 79.68 | 64.71 | 60.90 | 14.57 | 64.62 | 62.54 | 61.73 | 64.78 | 60.79 | 62.82 |
| FFN-Wider BERT w/ CAA (12.5%) | **75.48** | **83.33** | **81.66** | **65.00** | **62.23** | **16.49** | **66.08** | **62.86** | 62.03 | **65.14** | 61.37 | **63.79** |

Table 14: Fine-tuning results on the development set of SuperGLUE Benchmark of various BERT models (Pre-training Steps Alignment & Pre-training Performance Alignment).

| Model | HellaSwag | PIQA | WinoGrande | OpenBookQA | ARC Easy | ARC Chal. | Avg. |
|---|---|---|---|---|---|---|---|
| *(H=768, Pre-Training Performance≈ –1.53, Pre-Training Steps= 164.9k)* | | | | | | | |
| FFN-Wider BERT | 38.95 | 67.00 | 55.09 | **57.78** | 51.74 | 39.07 | 51.61 |
| FFN-Wider BERT w/ CAA (12.5%) | **40.59** | **67.81** | **57.14** | 57.70 | **51.94** | **39.49** | **52.45** |

Table 15: Fine-tuning results on multiple other tasks of various BERT models (Pre-training Steps Alignment & Pre-training Performance Alignment).

| | H=128 | | H=768 | |
|---|---|---|---|---|
| | FFN-Wider GPT | FFN-Wider GPT w/ CAA (37.5%) | FFN-Wider GPT | FFN-Wider GPT w/ CAA (37.5%) |
| Validation | -4.067 | -4.066 | -3.191 | -3.192 |
| ArXiv | -4.927 | -4.966 | -4.061 | -4.010 |
| BookCorpus2 | -4.055 | -4.054 | -3.315 | -3.309 |
| Books3 | -4.495 | -4.485 | -3.715 | -3.701 |
| DM Mathematics | -4.162 | -4.110 | -3.597 | -3.540 |
| Enron Emails | -4.636 | -4.598 | -3.877 | -3.852 |
| EuroParl | -6.026 | -5.995 | -4.855 | -4.865 |
| FreeLaw | -4.625 | -4.616 | -3.726 | -3.720 |
| Github | -5.030 | -4.980 | -4.189 | -4.156 |
| Gutenberg (PG-19) | -4.610 | -4.604 | -3.930 | -3.912 |
| HackerNews | -4.738 | -4.729 | -4.061 | -4.053 |
| NIH ExPorter | -4.617 | -4.611 | -3.657 | -3.642 |
| OpenSubtitles | -3.345 | -3.320 | -2.886 | -2.887 |
| OpenWebText2 | -4.817 | -4.810 | -3.975 | -3.964 |
| PhilPapers | -5.580 | -5.557 | -4.522 | -4.517 |
| Pile-CC | -4.758 | -4.754 | -3.979 | -3.968 |
| PubMed Abstracts | -4.591 | -4.576 | -3.593 | -3.579 |
| PubMed Central | -4.597 | -4.576 | -3.797 | -3.780 |
| Stack Exchange | -5.031 | -5.013 | -4.233 | -4.210 |
| Ubuntu IRC | -5.729 | -5.623 | -4.989 | -4.942 |
| USPTO Backgrounds | -4.503 | -4.468 | -3.690 | -3.668 |
| Wikipedia (en) | -4.312 | -4.295 | -3.359 | -3.354 |
| YoutubeSubtitles | -4.822 | -4.800 | -4.036 | -4.017 |
| Average | -4.728 | -4.706 | -3.911 | -3.893 |

Table 16: Out-of-distribution language modeling results on the development set of Pile of various GPT models (Pre-training Steps Alignment & Pre-training Performance Alignment).

| Model | HellaSwag | PIQA | WinoGrande | COPA | OpenBookQA | ARC Easy | ARC Chal. | StoryCloze | Winograd | Avg. |
|---|---|---|---|---|---|---|---|---|---|---|
| *(Random Performance)* | 25.00 | 50.00 | 50.00 | 50.00 | 25.00 | 25.00 | 25.00 | 50.00 | 50.00 | 38.89 |
| *(H=768, 0-Shot, Pre-Training Performance≈ –3.19, Pre-Training Steps= 50.9k)* | | | | | | | | | | |
| FFN-Wider GPT | 26.79 | **57.77** | 51.62 | 57.00 | 29.20 | **43.56** | 22.45 | 56.49 | 56.49 | 44.60 |
| FFN-Wider GPT w/ CAA (37.5%) | **26.83** | 56.78 | **52.09** | **58.00** | **30.60** | 42.68 | **25.17** | **57.35** | **57.90** | **45.27** |
| *(H=768, 1-Shot, Pre-Training Performance≈ –3.19, Pre-Training Steps= 50.9k)* | | | | | | | | | | |
| FFN-Wider GPT | 27.54 | **57.18** | 50.33 | 63.10 | 28.36 | **40.72** | 25.20 | 56.24 | 54.88 | 44.84 |
| FFN-Wider GPT w/ CAA (37.5%) | **28.04** | 56.57 | **51.03** | **63.30** | **30.72** | 40.34 | **25.95** | **57.16** | **59.23** | **45.82** |

Table 17: Zero-shot and one-shot results on multiple datasets of various GPT models (Pre-training Steps Alignment & Pre-training Performance Alignment).

