# OpenReview forum: "How does Architecture Influence the Base Capabilities of Pre-trained Language Models? A Case Study Based on FFN-Wider and MoE Transformers"
_NeurIPS.cc/2024/Conference — NeurIPS 2024 poster_

### Official Review · Reviewer_nte2 · 2024-07-08

**Soundness:** 3
**Presentation:** 3
**Contribution:** 3
**Rating:** 6
**Confidence:** 3

**Summary:**

This study investigates the influence of architecture on pre-trained language models' base capabilities. It reveals that the contribution ratio of Multi-Head Attention to pre-trained language modeling affects base capabilities. FFN-Wider Transformers reduce this ratio, leading to a decline in base capabilities. The researchers propose a Combination Enhanced Architecture (CEA) to address this issue. They also extend this to Mixture of Experts (MoE) transformers, achieving significant improvements in base capabilities.

**Strengths:**

+ The paper shifts the focus from the commonly studied impact of scale on pre-trained language models to the impact of architecture.
+ The exploration of the influence of using wider FFN layer is interesting, and should impact further researches.
+ The proposal of CEA as a solution to the decline in base capabilities shows a proactive approach to solving the identified problem.
+ The findings are confirmed through extensive experiments, adding credibility to the claims.

**Weaknesses:**

- The description of CEA is not detailed enough.

**Questions:**

In general, I think the discovery of the paper is interesting, and I don't have many questions. Just some minor issues:

- What is the definition of mutual information? Give some introduction.
- How is the pre-training performance of the models?
- How is the structure of CEA? It is not clear to me after reading Section 6.

**Limitations:**

The authors address it in Section 8.

---

> ### Author Rebuttal · Authors · 2024-08-07
>
> Thanks for your insightful review and valuable feedback!
>
> We answer your questions below.
>
> ***
>
> **Q1:** How is the pre-training performance of the models?
>
> **A1:** For the BERT and GPT experiments, the results are primarily presented in Tables 1 and 2, with a textual description of the pre-training performance achieved by each model group. For the MoE model experiments, **we have included in the attached PDF (in the global response above) pre-training performance curves similar to Figure 1(c).** These curves allow for a comparison between the original MoE model and the improved MoE model, which we hope addresses your concerns.
>
> ***
>
> **Q2:** What is the definition of mutual information? Give some introduction.
>
> **A2:** We apologize for the oversight. The absence of the definition of mutual information might cause confusion for researchers outside this field. We will follow your suggestion and add the relevant definition and explanation of mutual information.
>
> ***
>
> **Q3:** How is the structure of CEA? It is not clear to me after reading Section 6.
>
> **A3:** We apologize for any confusion caused, and we would like to provide further clarification.
>
> In Section 5, we introduce the Combination Adjustable Architecture (CAA), which is an architecture designed for analytical purposes. This architecture primarily involves splitting the original FFN in the FFN-Wider model according to a certain width ratio to obtain Outer-FFN and Inner-FFN. We then adjust the width ratio between these two components to support our arguments. At this stage, we pre-train multiple models with varying width ratios on a small scale for analysis.
>
> In Section 6, we introduce the Combination Enhanced Architecture (CEA), which is similar to the CAA mentioned above. However, since CEA is an improved architecture rather than an analytical one, it requires a fixed width ratio to be determined before conducting large-scale pre-training experiments. For different models, we determined various width ratios and then performed large-scale pre-training and downstream experiments.
>
> ***

---

> > ### Comment · Reviewer_nte2 · 2024-08-14
> >
> > Thank you for the rebuttal! It addresses my concerns and I will keep my rating.

---

### Official Review · Reviewer_QyvJ · 2024-07-09

**Soundness:** 3
**Presentation:** 3
**Contribution:** 3
**Rating:** 6
**Confidence:** 3

**Summary:**

The paper studies the contribution raion of FFN and MHA layer in transformers and its effect on out-of-distribution performance. It finds that the wider FFN layer will increase its contribution ratio and lower the OOD performance. Lastly, the paper proposes a new architecture that moves part of the FFN layer to MHA and shows the CEA can improve the baseline models' OOD ability.

**Strengths:**

- The paper analyzes the contribution ratio with new methods that evaluate Mutual Information and token prediction accuracy of each layer's output.

- The proposed CEA method has improved the MoE transformer in various tasks and datasets with the same parameter scale, showing its effectiveness in increasing OOD ability.

**Weaknesses:**

- The paper does not give a convincing explanation of why the study aligns pre-trained performance with different parameter scales. A larger scale model may not fully converge when it has a similar training loss to a smaller model. It is also not known whether the proposed CEA will harm in-domain performance.

- The MHA module in the vanilla transformer also has linear layers for Q, K, V, and after self-attention. I think they can be seen as inner-FFNs that will transform the combination function and make it not necessary to introduce an extra inner-FFN layer.

- Tables 1 and 2 show that BERT w/ CEA performs worse than vanilla in GLUE and SGLUE and GPT w/ CEA perform worse in all datasets.

**Questions:**

See weaknesses 1 and 2.

Others:

- Figure 6 lacks a legend to explain the figure elements, including the blue line, orange bar, blue and orange dotted lines.

**Limitations:**

The authors addressed the limitations.

---

> ### Author Rebuttal · Authors · 2024-08-07
>
> Thanks for your insightful review and valuable feedback!
>
> We answer your questions below.
>
> ***
>
> **Q1:** The paper does not give a convincing explanation of why the study aligns pre-trained performance with different parameter scales. A larger scale model may not fully converge when it has a similar training loss to a smaller model. It is also not known whether the proposed CEA will harm in-domain performance.
>
> **A1:** You mentioned why we adopted the alignment of pre-training performance for models with different parameter scales. We would like to explain this further. We mainly consider that comparing models with different parameter scales is often unavoidable (even outside the scope of this paper). Therefore, finding a relatively reasonable basis for cross-parameter scale comparison is necessary. **This method needs to eliminate the interference of parameter advantages, computational advantages, etc., and maximally reflect the base capability differences brought by the architecture itself.** In our paper, we explained that aligning pre-training performance might be more suitable for this goal (Section 2.2), while other methods might not be appropriate (Appendix A).
>
> You also mentioned the potential issue of inadequate convergence in large parameter models, and you might be concerned about the inconsistency in the behavior of models during the later stages of training. While this factor cannot be entirely ruled out, we have not yet explored this topic deeply in our current analysis. **However, one thing is certain:** in our work, the comparison between large parameter models and their improved versions does not face this issue, as they have the same parameters, and there should not be significant differences in the degree of convergence.
>
> Regarding your concern about whether the new architecture would harm in-domain performance, as shown in Figure 6, most width ratios do not pose any problems. Only extreme width ratios might harm pre-training performance. Our experiments have demonstrated that choosing non-extreme width ratios can still achieve performance gains (Appendix I). For MoE models, **we have plotted pre-training performance curves similar to Figure 1(c) in the attached PDF (in the global response above).** It can be seen that the improved MoE models do not experience a degradation in pre-training performance, which essentially indicates the usability of the new architecture.
>
> ***
>
> **Q2:** The MHA module in the vanilla transformer also has linear layers for Q, K, V, and after self-attention. I think they can be seen as inner-FFNs that will transform the combination function and make it not necessary to introduce an extra inner-FFN layer.
>
> **A2:** You mentioned that there are also linear layers within the MHA module, which can be considered as inner-FFN layers, making additional inner-FFN layers potentially unnecessary.
>
> We had considered this issue as well, but we found that the linear layers in the MHA module lack non-linear transformations, which directly results in their transformation capability being inferior to that of the FFN layers. Therefore, considering the linear layers in the MHA as inner-FFN might still face the issue of the MHA contribution ratio as discussed in our paper. Ultimately, we chose to convert a part of the FFN into inner-FFN.
>
> ***
>
> **Q3:** Tables 1 and 2 show that BERT w/ CEA performs worse than vanilla in GLUE and SGLUE and GPT w/ CEA perform worse in all datasets.
>
> **A3:** Your observations are correct, and they are reasonable within the context of our work.
>
> Firstly, since the models in Tables 1 and 2 are aligned based on pre-training performance, the FFN-Wider model cannot outperform the vanilla model solely based on parameter size. Therefore, it is expected that the vanilla model performs better on downstream tasks, and it is normal for the FFN-Wider model to be inferior to the vanilla model.
>
> Secondly, **the fact that FFN-Wider w/ CEA is also inferior to the vanilla model is within our expectations.** The main reason is that we used the FFN-Wider model primarily as a good subject for analysis. Improving it was only to demonstrate the effectiveness of our analysis, and ultimately, the improvements are intended to benefit the MoE model.
>
> In fact, the FFN-Wider model does not have efficiency advantages over the vanilla model. Improving it is less practical than directly using an enlarged vanilla model. Thus, even if the FFN-Wider w/ CEA were to surpass the vanilla model, it would not have practical significance. Therefore, we ultimately focus on demonstrating practical utility through the MoE model.
>
> ***
>
> **Other Responses:**
>
> We apologize for any inconvenience caused by the figures in the paper. We will make improvements in subsequent versions to enhance the clarity and comprehensiveness of the related expressions in the paper.
>
> ***

---

> > ### Comment · Reviewer_QyvJ · 2024-08-13
> >
> > Thanks for the response, it addresses most of my concerns.
> >
> > I still have some concerns about the results of BERT/GPT w/ CEA. I understand that the FFN-Wider performs worse than the vanilla model, but I think w/ CEA should at least achieve similar results to the vanilla model. Otherwise, there is no reason to use CEA rather than vanilla. Maybe authors can show that the CEA can outperform vanilla with the same parameter scale and not align pre-training performance.
> >
> > However, the results of MoE are convincing. I think this is a good paper and will raise my rating.

---

### Official Review · Reviewer_Rvop · 2024-07-13

**Soundness:** 2
**Presentation:** 2
**Contribution:** 3
**Rating:** 4
**Confidence:** 3

**Summary:**

This paper examines how the architecture of a transformer model influences its base capabilities, such as out-of-distribution tasks, transfer learning, and few-shot learning. Specifically, it explores the effects of replacing the feed-forward network (FFN) with a wider FFN (FFN-wide) in various parts of the architecture.

Initially, the authors find that replacing the FFN (referred to as the transformation function) after the attention layer (referred to as the combination function) with FFN-wide leads to worse performance. They analyze this impact by measuring the contribution of the combination function using techniques such as mutual information and token prediction. The results indicate that performance deteriorates in most cases when the contribution of the combination function decreases.

The authors then devise multiple architectures with different width ratios of FFN-wide in the combination and transformation functions. They select the best ratio model architecture, termed CEA (Combination-Enhanced Architecture). This CEA architecture is subsequently used in a Mixture of Experts (MoE) model, demonstrating improvements over the non-CEA MoE model.

**Strengths:**

- The paper addresses an interesting problem by measuring the impact of transformer architecture on the downstream performance of the model, which is crucial for optimizing and understanding transformer models.

- Attributing performance to the inner workings of a neural network, particularly the contribution of specific components, and analyzing this contribution using techniques like mutual information is both novel and insightful.

- It is also interesting to see the impact on two different kinds of model like Bert and GPT and see how the optimal width for the two types of models are different.

- The application of CEA to a mixture of experts models definitely demonstrates the practical benefits of this paper.

**Weaknesses:**

- The performance improvements for the different variations of the architectures seem minor, and the absence of standard deviations makes it difficult to assess the robustness of the results.

- For both BERT and GPT, even after a thorough architecture search, the performance is at best similar to the vanilla models, which is not very promising.

- Similarly for the MoE experiment, it will useful to add the standard deviation of the results. Additionally, since the training loss is lower for MoE with CEA compared to vanilla MoE, it is hard to determine whether the improvements are due to the architecture's inductive bias or just the lower training loss. Similar to the rest of the paper, please also report numbers for this experiments by keeps the same training performance level.

Overall, the paper is interesting, but the performance values being so close and the lack of variance measurement do not give me much confidence in the results.

Minor: In Table 1 and Table 2, the results for the proposed approach are bolded rather than the best approach for that experiment.

In most figures, the y-axis does not start at 0, which can give readers an inaccurate representation of the data.

**Questions:**

- What does it mean for the width ratio to be 0% and 100%? Does this imply that the FFN is removed in these cases?

- What does it mean that FFN-wider models with CEA are not able to beat vanilla models in some cases in Table 1 and Table 2? Isn't the vanilla model one of the ratios explored? If not, would it make more sense to explore the architectures on some other dimensions?

- Please indicate whether lower or higher values are better for all the tables and figures in the paper.

**Limitations:**

The authors can also address the limitations of exploring only a narrow subset of architectures and discuss whether any parts of the proposed methodology can be extended to other aspects of transformer architectures. Additionally, all experiments were performed on smaller-scale models, so the results may not generalize to larger models.

---

> ### Author Rebuttal · Authors · 2024-08-07
>
> Thanks for your insightful review and valuable feedback!
>
> We answer your questions below.
>
> ***
>
> **Q1:** since the training loss is lower for MoE with CEA compared to vanilla MoE, it is hard to determine whether the improvements are due to the architecture's inductive bias or just the lower training loss ... please also report numbers for this experiments by keeps the same training performance level.
>
> **A1:** **Our MoE experiments still utilized a similar setting: ensuring that the pre-training performance of the original MoE and the improved MoE were roughly consistent.** We suspect that some results in Table 3 or Figure 1(c) may have caused the misunderstanding, so we would like to clarify them here.
>
> In Table 3, only "SlimPajama-CC&C4 (Loss)" represents pre-training performance, as the models in Table 3 were pre-trained on the CC and C4 subsets of the SlimPajama dataset. Because the SlimPajama dataset achieved de-duplication across subsets, the remaining results for SlimPajama belong to OOD tests. The performance curves in Figure 1(c) are similar, only representing OOD performance.
>
> Another concern might be that even if "SlimPajama-CC&C4 (Loss)" in Table 3 represents pre-training performance, the improved MoE (2.303) still seems to perform better than the original MoE (2.315). Is this advantage small enough? To illustrate the issue clearly, **we have included a PDF attachment (in the global response above)** with pre-training performance curves similar to Figure 1(c) and compared side-by-side with the original Figure 1(c).
>
> **It can be seen that:** 1) The gap in pre-training performance is negligible compared to the OOD performance gap; 2) During pre-training, there were moments when the pre-training performance of both models was identical, yet the OOD performance gap remained significant. Therefore, we believe this situation still largely aligns with our claimed setup.
>
> ***
>
> **Q2:** For both BERT and GPT, even after a thorough architecture search, the performance is at best similar to the vanilla models, which is not very promising.
>
> **A2:** For the FFN-Wider model, it is indeed the case that it does not offer efficiency advantages over the vanilla model. Improving the FFN-Wider model is less beneficial than simply using a scaled-up vanilla model. In fact, we did not emphasize its practical value but rather used it as a good analytical object to conduct some meaningful analysis.
>
> The key point is that we eventually extended the relevant analysis to the MoE model. The MoE model is a practical architecture that offers the advantage of expanding model capacity with lower computational costs compared to the vanilla model. With the same amount of computation, the MoE model performs better in pre-training, making it worthwhile to improve the MoE model.
>
> ***
>
> **Q3:** What does it mean that FFN-wider models with CEA are not able to beat vanilla models in some cases in Table 1 and Table 2? Isn't the vanilla model one of the ratios explored? ...
>
> **A3:** The response to A2 should also apply to this question.
>
> Additionally, it is necessary to clarify that the FFN-Wider model has a wider FFN, and even after adjusting width ratios, it cannot be equivalent to the vanilla model.
>
> ***
>
> **Q4:** What does it mean for the width ratio to be 0% and 100%? ...
>
> **A4:** Generally, adjusting the width ratio results in the presence of two FFNs. However, when the ratio is 0% or 100%, one of the FFNs is removed, leaving only a single FFN.
>
> ***
>
> **Q5:** Please indicate whether lower or higher values are better for all the tables and figures in the paper.
>
> **A5:** Our explanation is as follows:
>
> **Figure 1(b):** The orange line represents a metric, where higher values are better.
> **Figure 1(c):** Lower values are better.
> **Figure 2:** Higher values are better.
> **Figures 3 and 4:** No metrics are present.
> **Figure 6:** The blue line represents a metric, where higher values are better.
> **Tables 1 and 2:** Higher values are better.
> **Table 3:** Metrics labeled with "Loss" and "PPL" are better when lower, while metrics labeled with "Acc." are better when higher.
>
> The tables in the appendix provide detailed information, with the metrics having similar meanings.
>
> ***
>
> **Q6:** The performance improvements for the different variations of the architectures seem minor, and the absence of standard deviations makes it difficult to assess the robustness of the results ...
>
> **A6:** We would like to address this from two perspectives. **On one hand,** similar to A2 above, we do not actually consider the FFN-Wider model as a replacement for the vanilla model. The FFN-Wider model does not have practical competitiveness; its performance improvement is mainly to validate our analysis. Therefore, the extent of the improvement may not be particularly critical, and our main focus is to show the performance of the MoE model. **On the other hand,** given that there were no changes in scale or data, only slight architectural modifications, the extent of this performance improvement seems reasonable. Although it has limited practical significance, it generally supports our analysis.
>
> As for the robustness, on one hand, the results in Tables 1 and 2 are averaged performance across multiple tasks, which provides some degree of robustness. On the other hand, **we have supplemented the results with standard deviations in the PDF attachment (in the global response above)**, also indicating that the results are relatively robust.
>
> The tables in the PDF attachment only list the results involving multiple rounds of experiments. Specifically, the results in Table 1 are from 8 rounds (H=128) and 4 rounds (H=768) of experiments, Table 2 from 10 rounds, and Table 3 from 5 rounds.
>
> ***
>
> **Other Responses:**
>
> We apologize for any inconvenience caused by the tables and figures. We will make improvements in subsequent versions to enhance the clarity and comprehensiveness of the related expressions.
>
> ***

---

> ### Author Response · Authors · 2024-08-14
>
> Thank you very much for reviewing our work!
>
> We have addressed some of the concerns you raised in our rebuttal. Specifically, for the issues you were particularly concerned about, such as the fairness of the MoE experiment comparisons and the standard deviation of the experimental results, we have provided more detailed results (in the global response PDF above). We strongly believe that these additional results will effectively address your concerns, especially regarding the fairness of the MoE experiment comparisons, which seems to have led to some misunderstanding.
>
> If time permits, we kindly ask you to review our rebuttal. We look forward to your feedback. Thank you!

---

### Official Review · Reviewer_xAuN · 2024-07-13

**Soundness:** 3
**Presentation:** 2
**Contribution:** 3
**Rating:** 7
**Confidence:** 4

**Summary:**

The paper examines the influence of architecture on the base capabilities (OOD, transfer learning, few-short learning) of large language models. The main focus is on FFN-Wider transformers and understanding why they have poorer base capabilities compared to vanilla transformers. The contibution ratio of multihead attention to pretrained language modeling is found to be a key factor affective base capabilities. Based on this observation, a Combination Enhanced Architecture is proposed and also extended to mixture of experts transformers. The analytical study is backed by experimental evaluation successfully achieving significant improvements in base capabilities of the 14B parameter MoE model.

**Strengths:**

+ While most existing study of large models has focussed on the impact of scale, this work is a significant effort in understanding the influence of architecture.

+ The empirical findings are supported by a more deeper interpretation of the underlying mechanisms of these influences.

+ The proposed CAA wherein the wider FFN is split into adjustable two parts - outer-FFN which stays as a transformation function, and an inner-FFN which is relocated within the MHA layer for enhancing the combination function is interesting in its own right.

+ The extension to MoE transformers and the experimental demonstration with 14B parameter GPT architecture MoE model is very interesting.

**Weaknesses:**

- Some of the findings appear to be obvious (perhaps, in hindsight) - as the contribution ratio of the MHA layer increases, the base capabilities also improve. It would have been interesting to also find some non-intuitive influences.

- Overall the paper's analysis is much narrow in scope than the title of the paper. Nonetheless, it is interesting and useful study.

- The language and presentation of the paper can be improved. It will be useful to have the grammar issues fixed in the paper before the final version.

**Questions:**

- Some choices of parameters such as chosing intermediate dimension to 32d instead of say 8 or 16d can be better explained.

- Was the width adjustment limited to Outer-FFN?

**Limitations:**

The reviewer does not except any negative societal impact of this work.

---

> ### Author Rebuttal · Authors · 2024-08-07
>
> Thanks for your insightful review and valuable feedback!
>
> We answer your questions below.
>
> ***
>
> **Q1:** Some choices of parameters such as chosing intermediate dimension to 32d instead of say 8 or 16d can be better explained.
>
> **A1:** We agree with your opinion that attempting more width would enhance the persuasiveness of the work. In fact, the choice of 32d in the paper was not made with any special consideration; it was simply based on an assessment of computational resources. In future work, we will consider enriching this aspect to strengthen the persuasiveness of the work.
>
> ***
>
> **Q2:** Was the width adjustment limited to Outer-FFN?
>
> **A2:** Adjusting the width ratio is mainly aimed at the original FFN in the FFN-Wider model, which will be split into Outer-FFN and Inner-FFN. The sum of their widths is equal to the original FFN, meaning an increase in the width of one will result in a decrease in the width of the other. Therefore, the width adjustment should occur simultaneously in both the Outer-FFN and the Inner-FFN.
>
> ***
>
> **Other Responses:**
>
> We sincerely apologize for any inconvenience caused by the language and expression issues in the paper. We will improve and refine the relevant expressions in the subsequent versions of the paper.
>
> ***

---

> > ### Comment · Reviewer_xAuN · 2024-08-13
> > **Thank you**
> >
> > Thank you for addressing my concerns. Having carefully read all other reviews and your responses, I will keep my positive score.

---

### Author Rebuttal · Authors · 2024-08-07

Here is a PDF attachment containing the figures and tables referenced in the detailed responses below.

---

### Decision · Program_Chairs · 2024-09-25

**Decision:**

Accept (poster)

**Comment:**

This paper explores how and why the FFN-Wider architecture influences base capabilities of LLMs (out-of-distribution modeling, transfer learning, and few-shot learning). Specifically, when the pretraining performance is held constant, FFN-Wider Transformers have worse base capabilities than vanilla ones, but why? The authors identify a property of the architecture (the contribution ratio of MHA layer) that correlates with base capability, and provide further analysis and insights. The authors then propose a specific architecture (with the best ratio) which is shown to improve base capabilities in MoE models.

I recommend to **accept** this paper. Reviewers appreciated that the authors chose to study the influence of architecture (instead of scale) on LLM performance, which is often difficult due to the low interpretability of neural networks in general, but nevertheless the authors provided "novel and insightful" analysis into the topic. It is also significant and impactful that the authors found a better (than vanilla) architecture for MoE models, based on that earlier analysis.

However, reviewers also mentioned that the writing/clarity of the paper could be improved. I am optimistic about this aspect because the authors clarified reviewers' questions in the rebuttals, and I encourage the authors to revise the paper to include the clarifications and generally improve the presentation for the next revision.